# Bioremediation of Hexavalent Chromium by Chromium Resistant Bacteria Reduces Phytotoxicity

**DOI:** 10.3390/ijerph17176013

**Published:** 2020-08-19

**Authors:** Shanewaz Hossan, Saddam Hossain, Mohammad Rafiqul Islam, Mir Himayet Kabir, Sobur Ali, Md Shafiqul Islam, Khan Mohammad Imran, M. Moniruzzaman, Taslin Jahan Mou, Anowar Khasru Parvez, Zahid Hayat Mahmud

**Affiliations:** 1Laboratory of Environmental Health, Laboratory Sciences and Services Division, International Centre for Diarrhoeal Disease Research, Bangladesh, Dhaka 1212, Bangladesh; shanewaz.hossain@icddrb.org (S.H.); saddamimrd@bcsir.gov.bd (S.H.); mrafiqul@icddrb.org (M.R.I.); mkf37@umsystem.edu (M.H.K.); sobur.ali@icddrb.org (S.A.); mshafiq@icddrb.org (M.S.I.); imrankhan@vt.edu (K.M.I.); moniruzzaman1@icddrb.org (M.M.); 2Department of Microbiology, Jahangirnagar University, Savar, Dhaka 1342, Bangladesh; moumicro@juniv.edu (T.J.M.); khasru73@juniv.edu (A.K.P.); 3Industrial Microbiology Research Division, Bangladesh Council of Scientific and Industrial Research, Chittagong 4220, Bangladesh

**Keywords:** hexavalent chromium, chromium resistant bacteria, biosorption, isotherm, phytotoxicity

## Abstract

Chromium (Cr) (VI) has long been known as an environmental hazard that can be reduced from aqueous solutions through bioremediation by living cells. In this study, we investigated the efficiency of reduction and biosorption of Cr(VI) by chromate resistant bacteria isolated from tannery effluent. From 28 screened Cr(VI) resistant isolates, selected bacterial strain SH-1 was identified as *Klebsiella* sp. via 16S rRNA sequencing. In Luria–Bertani broth, the relative reduction level of Cr(VI) was 95%, but in tannery effluent, it was 63.08% after 72 h of incubation. The cell-free extract of SH-1 showed a 72.2% reduction of Cr(VI), which indicated a higher activity of Cr(VI) reducing enzyme than the control. Live and dead biomass of SH-1 adsorbed 51.25 mg and 29.03 mg Cr(VI) per gram of dry weight, respectively. Two adsorption isotherm models—Langmuir and Freundlich—were used for the illustration of Cr(VI) biosorption using SH-1 live biomass. Scanning electron microscopy (SEM) analysis showed an increased cell size of the treated biomass when compared to the controlled biomass, which supports the adsorption of reduced Cr on the biomass cell surface. Fourier-transform infrared analysis indicated that Cr(VI) had an effect on bacterial biomass, including quantitative and structural modifications. Moreover, the chickpea seed germination study showed beneficial environmental effects that suggest possible application of the isolate for the bioremediation of toxic Cr(VI).

## 1. Introduction

Wastewater from industrial applications contain potentially toxic metals and anonymous organics that cause alarming environmental pollution worldwide [1,2]. Despite chromium (Cr) being a highly toxic metal, different tannery industries frequently use this metal [3]. Chromium discharge in wastewater contaminates water bodies, endangers aquatic lives, and poses a severe health hazard [4]. Hence, it is a public health concern that needs to be addressed by reducing toxicity or Cr load from industrial effluents [3]. 

There are about 250 leather tanning industries at Hazaribagh and Hemayetpur in Dhaka city, the capital of Bangladesh, which discharge liquid and solid wastes into canals and rivers, thus increasing the amount of Cr in the surrounding water bodies [5]. Tannery industries use chrome salts (Cr(III)) as primary tanning agents during leather processing, but under suitable conditions, it can be readily oxidized to Cr(VI), where the later is more toxic than the former [3,6,7,8]. Human Rights Watch reported that the tannery wastewater from 47 tanneries in Hazaribagh contained extremely elevated levels of total chromium (4043 mg/L). On top of that, the tannery effluents in Chittagong, the second biggest city in Bangladesh, contained hexavalent chromium at 70.33 mg/L [9]. Cr(VI) was also found in deep tube wells in the Hazaribagh area, which exceeded the Bangladesh standard limit (0.05 mg/L) of chromium in drinking water [2].

As Cr(VI) is toxic and highly soluble in water, it can pass rapidly through cell membranes and eventually interact with proteins and nucleic acids [10]. Thus, the accumulation of toxic heavy metals in humans leads to carcinogenesis, mental retardation, renal malfunction, and other abnormalities [11]. The prevalence of diseases such as scabies (73.9%), gastrointestinal problem (71.7%), diarrhea (71.7%), asthma (49.9%), eye problems (46.7%), and high blood pressure (52.2%) was reported earlier among tannery workers in Bangladesh [12]. 

Therefore, it is of the utmost importance to treat these effluents they are discharged into the environment in order to reduce environmental pollution and the disease burden [3,6]. Numerous technologies are available to mitigate Cr contaminated wastewater, which includes redox chemical processes [13], reverse osmosis, coagulation, and precipitation [14,15]. However, these aforementioned methods are expensive and energy-consuming, and they have the risk of secondary chemical contamination [13,16]. A recent study mentioned that magnetically-assisted chemical separation techniques that use potential nanoadsorbents based on core-shell biomagnetic nanoparticles can remove Cr(VI) from aqueous solutions [17]. Wastewater and effluents with heavy metal contamination can alternatively be treated with potential microorganisms. These microorganisms offer large volumes of effluent treatments with a low operational cost and low energy demand with higher efficiency of metal removal [16,18,19]. Microorganisms provide higher surface area to the volume due to the small size and can assimilate metals from surrounding settings [20]. Various microorganisms are found in the industrial effluents discharging area and are capable of protecting themselves from the toxicity of existing heavy metals in the effluents [21,22,23,24]. These microorganisms including bacteria, fungi, algae, and protozoa use diverse systems for survival against heavy metal toxicity, such as uptake of heavy metal, adsorption, oxidation, methylation, and reduction of heavy metals to nontoxic forms [21,25]. Toxic Cr(VI) reduction to its nontoxic Cr(III) form is one of the mechanisms used by many organisms to survive in Cr(VI) contaminated effluents [24]. These microorganisms include *Acinetobacter* and *Ochrobactrum* [26]; *Arthrobacter* [27]; *Serratia marcescens* [28]; *Bacillus* spp. [29]; *Cellulomonas* spp. [30]; *Pseudomonas fluorescens* LB300; *Intrasporangium* sp. Q5-1; *Enterobacter cloacae; Bacillus* sp. ES29; and *E. coli* [31,32,33,34,35]. Moreover, a mixture of different microbes [36] can reduce the mobility and toxicity of Cr(VI) by transforming to a less mobile and less toxic form of Cr(III). 

The application of microbes for bioremediation of heavy metals and other contaminants has been studied in other countries [37] but yet to be initiated in a developing country like Bangladesh. There is no detailed study on the application of chromium resistant bacteria for treating chrome-polluted wastewater. Therefore, the present study was conducted for the isolation and identification of Cr(VI) resistant bacteria from tannery effluents and their potential application in the removal of Cr(VI) from effluent water.

## 2. Materials and Methods

### 2.1. Materials

Chemicals and reagents used in this study were American Chemical Society (ACS) analytical grade unless stated otherwise. American Society for Testing and Materials (ASTM) type 1 deionized water (resistivity of 18.2 MΩ-cm) was used for all experiments supplied from Ultrapure Water Purification Systems (Arium comfort 2, Sartorius, Germany). Next, 1,5-diphenyl carbazide, NADH, K_2_Cr_2_O_7,_ NiCl_2_.6H_2_O, Pb(NO_3_)_2_, Co(NO_3_).6H_2_O, ZnCl_2_, CdCl_2_, and CuSO_4_.5H_2_O were obtained from Sigma-Aldrich (St. Louis, MO, USA). MnCl_2_ was purchased from Fisher Scientific (Waltham, MA, USA). All stock solutions were stored in amber glass bottles in a dark environment. Chickpea seeds were obtained commercially from the retail market in the Dhaka city, Bangladesh.

### 2.2. Sampling Sites and Sample Collection

The sampling site was selected based on the specific industrial area, where approximately 250 tannery industries were located [5]. In this study, tannery effluent samples were collected from different points in the Hazaribagh and Hemayetpur tannery industrial area of Dhaka metropolitan, Bangladesh, after obtaining appropriate permission from the authorities (Appendix A).

Tannery effluents from 14 tannery industries were taken in wide-mouth (Nalgene, Rochester, NY, USA) sterilized bottles from the nearby drain, close to the discharging outlet, from October to November 2017. In total, 500 mL of an effluent sample was collected for isolation and characterization of Cr(VI) resistant and reducing bacteria. We followed standard methods for collecting, processing, and measuring chemical and physicochemical parameters [38]. Physicochemical parameters (e.g., temperature, pH, salinity, electrical conductivity, total dissolved solids) were measured by a portable pH meter (Orion- 2 STAR, Thermo Scientific, Waltham, MA, USA) and HACH conductivity meter (SensION5, Loveland, CO, USA) at the sampling points. An atomic absorption spectrometer (AAS, Thermo Scientific, Waltham, MA, USA) determined the total chromium in the samples. The recorded temperature of the collected samples ranged from 18.4 to 26.1 °C; pH from 3.61 to 9.51; conductivity from 752 to 106,300 µS/cm; total dissolved solids (TDS) from 376 to 68,000 mg/L; and chloride from 68.86 to 4366.25 mg/L (Appendix A). Salinity and turbidity ranged from 0.4 to 74.5 ppt and 36.1 to 519 NTU, respectively. The COD of the collected samples ranged from 361 to 7788 mg/L (Appendix A). The total chromium load of the tannery effluents varied from 0.005 to 1423.05 mg/L (Appendix A). Guidelines for Bangladesh standard for wastewater disposal for these parameters are listed in Appendix A. 

### 2.3. Screening of Potential Chromium Reducing Bacteria

We screened for Cr(VI) resistance and reduced bacteria following the procedure shown in Figure 1a. For the isolation of chromium resistant bacteria, 50 µL of 14 tannery effluents were separately spread on Luria–Bertani (LB) agar plates supplemented with Cr(VI) concentration of 100 mg/L and incubated at 37 °C for 24 h. After this incubation period, the presence of microbial growth was observed and colonies were selected based on discrete morphological characteristics (Appendix A). Next, selected colonies were inoculated in LB broth media supplemented with increasing concentration of Cr(VI) from 500 mg/L to 2500 mg/L and incubated for 24 h at 37 °C in a rotary shaker incubator. Then, 50 µL of the cultures was dropped on nutrient agar (NA) media and incubated at 37 °C for 24 h. Colonies that survived at concentrations above 500 mg/L of Cr(VI) were selected from the NA plates and preserved for further study. As shown in Figure 1b, nine isolates were found to be resistant to more than 500 mg/L Cr(VI). We also screened for Cr(VI) sensitive bacteria from different species that were preserved in the Laboratory of Environmental Health to use as a negative control. An environmental isolate of *Vibrio cholerae* was found to be sensitive (can tolerate up to 100 mg/L Cr(VI); Figure 1b). One of the isolates (SH-1 can tolerate up to 2000 mg/L) showed very high resistance to Cr(VI) (Figure 1c). Next, Cr reduction study of the selected isolates was carried out in the LB broth medium containing 100 mg/L of Cr(VI). After 72 h of incubation at 37 °C in a rotary shaker (120 rpm), 1 mL from the respective cultures was taken, spun down at 11,000× *g* for 5 min, and supernatants were used to estimate remaining Cr(VI) concentration in the medium. Interestingly, SH-1 showed the highest rate (97%) of reduction (Figure 1d). After performing a one-way analysis of variance (ANOVA) followed by a post-hoc Tukey test, the isolates showed significant differences in the reduction of Cr(VI) compared to the negative control (*Vibrio cholerae*). The primary screening for Cr(VI) resistant bacteria showed that the organism with higher resistance had a higher reduction capacity. This intriguing finding leads to studying the mechanism of the reduction and possible application for bioremediation.

### 2.4. Molecular Identification and Phylogeny Analysis of the Selected Isolate

The selected bacterial isolate was morphologically characterized, based on microscopic observation. Then, the bacterial genomic DNA was extracted according to the manufacturer’s instruction using Maxwell 16 automated DNA extractor (Promega, Madison, WI, USA). The 16S rRNA was amplified by conventional Polymerase Chain Reaction (PCR) using a universal primer set for bacteria: 27F (5′-AGATTTGATCMTGGCTCAG-3′) and 1492R (5′-TACGGYTACCTTGTT ACGACTT-3′) [39]. The amplification was performed in a thermal cycler (G2 Gene Atlas, Astec, Fukuoka, Japan) consisting of 30 cycles. The PCR condition was optimized by maintaining an initial denaturation at 95 °C for 5 min, subsequent denaturation at 95 °C for 30 s, annealing at 48 °C for 30 s, extension at 72 °C for 90 s, and final extension for 5 min at 72 °C. The amplicons were then purified using Wizard^®^ SV Gel and a PCR clean-up system (Promega, Madison, WI, USA) and used for Sanger dideoxy sequencing. The sequencing was performed in 1st Base Laboratories (Malaysia) and the generated sequence was submitted to the Genbank/National Center for Biotechnology Information (NCBI) database. BLAST analysis was conducted to identify close phylogenetic relatives to identify isolated bacterium [40]. Then, the sequence alignment and the phylogeny analysis were performed using the MEGA software tool version 6.0 [41].

### 2.5. Determination of Optimum Growth Condition of the Isolate

The optimum temperature and pH for the growth of the isolate was determined according to the procedure described elsewhere [21]. The effect of Cr(VI) on the growth of the bacterial isolate was also investigated, according to Zahoor and Rehman (2009) [21]. Growth curves of the isolate SH-1 were determined from LB broth medium with 50 mg/L concentration of Cr(VI) and without chromium as the control. Here, an environmental isolate of *Vibrio cholerae* was used as a Cr(VI) sensitive organism (negative control). For each bacterial isolate, a 10 mL medium was taken in a sterilized falcon tube and inoculated with 20 µL of overnight grown inoculum. The cultures were incubated at 37 °C in a shaker at 120 rpm. An aliquot of culture was taken out at regular intervals from 0 h to 24 h. Absorbance was measured at 600 nm.

### 2.6. Resistance to Antibiotics and other Heavy Metals

Antibiotic susceptibility testing (AST) of the isolate was performed using the VITEK-2 system with VITEK 2 cards (AST-N280) for 19 antimicrobial agents according to the manufacturer’s recommendations and two additional antimicrobial agents (cefixime and ceftazidime) were incorporated. Then the Vitek susceptibility results were concluded according to the guidelines of the Clinical and Laboratory Standards Institute (CLSI). *E. coli* ATCC 25922 susceptible to all drugs were used for AST in each Vitek testing step for quality control. The 21 antibiotics tested included amikacin, amoxicillin/clavulanic acid, ampicillin, cefepime, cefixime, cefoperazone/Sulbactam, ceftazidime, ceftriaxone, cefuroxime, cefuroxime axetil, ciprofloxacin, colistin, ertapenem, gentamicin, imipenem, meropenem, nalidixic acid, nitrofurantoin, piperacillin, tigecycline, and trimethoprim/sulfamethoxazole. Minimum inhibitory concentration (MIC) of the isolate was interpreted as susceptible, moderate, or resistant based on CLSI guidance requirements. Isolate SH-1 was resistant only to ampicillin and sensitive to all other groups of antibiotics (Appendix A).

We prepared stock solutions of different metal salts containing 5000 mg/L concentration (i.e., lead nitrate, cadmium chloride, copper sulfate, zinc chloride, cobalt nitrate, manganese chloride, and nickel chloride) and increased the concentration (50 to 2000 mg/L) of the respective metals to determine the heavy metal resistance of bacterial isolate SH-1. Sterilized falcon tubes containing 10 mL LB medium with increasing concentration of respective metal ions were inoculated with 20 μL young bacterial cultures subsequently incubated in a rotary shaker incubator at 37 °C for 48 h. Then, droplets of 100 μL from respective test tubes were inoculated onto nutrient agar (NA) medium and incubated at 37 °C for 24 h. After this incubation period, the presence or absence of microbial growth concerning their heavy metal resistance ability was determined. The resistance pattern for heavy metals of the isolate was Cr^6+^ > Pb^2+^ > Mn^2+^ > Ni^2+^, Cu^2+^, Co^2+^ > Zn^2+^, Cd^2+^ (Figure 1e). 

### 2.7. Analysis of Reduction Capacity of Cr(VI) by the Isolate in the Culture Medium, Pond Water, and Tannery Effluents

The ability of bacterial isolate SH-1 to reduce Cr(VI) was examined in a sterilized 125 mL Erlenmeyer conical flask. LB broth media (25 mL) with added Cr(VI) concentration of 100 mg/L as K_2_Cr_2_O_7_ was inoculated with 20 μL of the selected young bacterial suspension. Cultures were incubated at 37 °C with 120 rpm. Then, 1 mL of the culture was taken after 24, 48, and 72 h. It was then centrifuged at 11,000× *g* for 5 min and the supernatant was analyzed for the residual Cr(VI) using the 1,5-diphenyl carbazide method [21]. To verify the performance of the isolate, the minimal salt broth was used following the above conditions.

The efficiency of the bacterial isolate to reduce Cr(VI) in LB broth, as well as in surface water containing Cr(VI), were tested according to the method described elsewhere [42]. In this experiment, 25 mL pond water and tannery effluents were taken separately in a sterilized 125 mL Erlenmeyer conical flask with added Cr(VI) concentration of 100 mg/L. Then, 20 μL of an overnight culture of the selected isolate was inoculated into the respective pond water and effluents. After incubation of 24, 48, and 72 h in an orbital shaker at 37 °C, samples (1 mL) were taken for the measurement of residual Cr(VI) in the solution using the above method [21].

### 2.8. Enzyme Assay

The bacterial isolates (SH-1 and *V. cholerae*) were grown in 10 mL LB broth for 48 h at 37 °C in a rotary shaker incubator with chromium (10 mg/L) and without chromium. Cell harvesting was performed by centrifugation at 11,000× *g* for 5 min. Pellets were individually washed twice with 10 mM phosphate buffer (pH 7.2) and stored at −20 °C. Then, B-PER bacterial protein extraction reagent with lysozyme and DNase 1 (Thermo Scientific, Rockford, IL, USA) were used for cell lysis on ice. The remaining lysates were centrifuged at 11,000× *g* for 5 min at 4 °C and the supernatants were collected separately to use as a crude cell extract. From the crude extracts, protein concentrations were measured using the modified Lowry Protein Assay Kit (Thermo Scientific, Rockford, IL, USA). Chromate reductase activity of this soluble proteins was examined as described elsewhere [43]. A reaction mixture of 1 mL was prepared containing 10 mg/L chromate in 0.85 mL of 10 mM phosphate buffer (pH 7.2). Then, 0.15 mL protein samples with NADH (14.3 μL from 140 μM) were added to the reaction mixture. In this experiment, protein samples, heated at 100 °C for 30 min, were used as blank control, and protein samples of *V. cholerae* acted as a negative control. After 12 h of incubation at 37 °C, Cr(VI) reduction was assessed according to the 1,5 diphenyl carbazide method [21].

### 2.9. Batch Biosorption Experiments

The preparation of live and dead cell dried biomass was performed as described earlier [44,45]. A batch equilibrium method [46] was used to determine the sorption of Cr(VI) by the dried biomass of the isolates (SH-1 live and SH-1 dead). All sets of experiments were done with 10 mg of the dried biomass in fixed volume (10 mL) of Cr(VI) solution in a sterilized falcon tube (15 mL). Bacterial biomass was exposed to Cr(VI) solution separately for 6 h on a rotary shaker (140 rpm) at 30 °C. The effects of pH, contact time, and initial Cr(VI) concentration on biosorption was recorded. All pH values of the solutions were adjusted from 1.0 to 8.0 using 1N HCl and 1N NaOH. The samples were taken at different time intervals and centrifuged at 11,000× *g* for 5 min afterward. The supernatant was analyzed for Cr(VI) concentration using the 1,5 diphenyl carbazide method [21]. 

#### 2.9.1. Analysis of Cr(VI) and Measurement of Metal Uptake

The 1,5-diphenyl carbazide method using a UV-vis Spectrophotometer (Varian, carry50, Walnut Creek, CA, USA), was used to determine the initial and final concentration of Cr(VI) following the procedure described elsewhere [21]. The formulae described by Vanderborght and Van Griekenm was used to calculate the amount of adsorbed Cr(VI) (mg/g) [47], which is as follows:(1)$$Q=\frac{V\;\left(C_{i}-C_{f}\right)}{M}$$
where, *Q* = capacity of Cr ion uptake (mg/g), *V* = adsorbate volume (L), *C_i_* = the initial concentration of Cr in solution before the sorption analysis (mg/L), *C_f_* = final concentration of Cr in solution after the sorption analysis (mg/L), and *M* = Dry weight of biosorbent (g). 

#### 2.9.2. Adsorption Isotherms 

Adsorption study with two of the most popular isotherm models—Langmuir and Freundlich—were carried out by mixing 10 mg of biosorbents with various concentrations of Cr(VI) ion solution ranging from 50 mg/L to 200 mg/L under the predetermined parameters above [48,49,50,51]. The Langmuir adsorption Equation (2) is as follows:(2)$$Q_{e}=\frac{Q_{o}K_{L}C_{e}}{1+K_{L}C_{e}}$$

The parameters of the Langmuir adsorption model were transformed into the following form:(3)$$\frac{1}{Q_{e}}=\frac{1}{Q_{o}}+\frac{1}{Q_{o}K_{L}C_{e}}$$

The equilibrium parameter, *R_L_*, and the separation factor are a dimensionless constant in Langmuir isotherm used to describe the adsorption nature of Cr(VI) [49].
(4)$$R_{L}=\frac{1}{1+\left(1+K_{L}C_{o}\right)}$$

The Freundlich model is suitable for the heterogeneous surface of sorbents to define the adsorption characteristics [48]: Q_e_ = K_f_ C_e_^1/n^(5)
(6)$${\mathrm{logQ}}_{e}=\log\;K_{f}+\frac{1}{n}\;{\mathrm{logC}}_{e}$$

K_f_ indicates approximate adsorption capacity; $\frac{1}{n}$ is a function of the strength of adsorption [52].

#### 2.9.3. Kinetics Studies

The kinetic study suggests the efficiency of the biosorbent and also explains the characteristics of the adsorption process for its effective application [53]. A kinetic study was carried out in batch and the adsorption of Cr(VI) onto biosorbents as a function of time with considering pseudo-first order Equation (7) and pseudo-second order Equation (8) rate equations [54].
log (q_e_ − q_t_) = log q_e_ − (K_1_/2.3030) t(7)
1/q_t_ = 1/K_2_ q_e_^2^ + (1/q_e_) t(8)
where K_1_ (min^−1^) and K_2_ (g/mg min) = rate constants of the respective kinetic models q_e_ = sorption capacities at equilibrium (mg/g) q_t_ = sorption capacities at time t (mg/g).

### 2.10. Scanning Electron Microscopic (SEM) and Fourier Transform Infrared Spectrometric (FTIR) Analysis

SEM and energy dispersive X-Ray (EDX) analyses were conducted to analyze the effect of Cr(VI) on bacterial biomass morphology. The bacterial biomass was mixed with 10 mL volume of Cr(VI) solution (100 mg/L) at pH 2.0. After 6 h of absorption, pellets were separated and washed twice with double deionized water. Then, the pellets were dried with a critical point dryer (CPD) and platinum-coated ion sputter coater (JEC-3000FC Auto Fine Coater, JEOL, Tokyo, Japan), and examined under a SEM (JEOL JSM-7610F) to observe the morphological changes in the bacterial biomass after biosorption. Bacterial dried biomass without subjected to Cr(VI) was used as the control. 

The functional groups on the cell surface responsible for the adsorption of chromium was identified by Fourier transform infrared spectrometric (FTIR) analysis. The bacterial biomass was mixed with 10 mL of Cr(VI) solution (100 mg/L) at pH 2.0. After 6 h of absorption, pellets were separated and washed twice with double deionized water. Then, the pellets were dried with air oven drier. Dried Cr(VI) treated biomass and untreated biomass were grounded with salts (KBr) separately by mortar and pestle. The resultant powder was converted into a 7 mm die set separation disc by manual pressing. The absorbance of the IR spectrum was recorded within 500 to 4000 cm^−1^ [55] using a FTIR (Alpha-II, BRUKER, Osann Monzel, Germany) at the Bangladesh Council of Scientific and Industrial Research (BCSIR) Laboratories, Chittagong, Bangladesh.

### 2.11. Assessment of Toxicity of Cr(VI) on Chickpea Seed Germination

For each group, three chickpea seeds were used and fungal contamination was avoided by immersing seeds into a 3% (*v/v*) formaldehyde solution for 5 min. Seeds were then washed with double deionized water followed by placing in a petri dish containing filter-sterile pond water with different concentrations of Cr(VI) (12.5, 25, 50, and 100 mg/L) and without Cr(VI) as the control. Seeds were set under 12/12 h light/dark cycle and temperature of around 25 °C during the day and around 18 °C during the night. The growth of seedlings was observed and measured after one week. 

To determine the Cr(VI) reducing effect, 50 mg/L of Cr(VI) were treated separately with the isolate SH-1 and dried SH-1 live biomass and then added into the seed germination plates described above. The filter-sterile pond water was used as a control, and pond water with 50 mg/L Cr(VI) was used as a negative control. Seeds were placed under the same conditions described above. Germination rate, as well as root and shoot length, were observed and measured after one week.

### 2.12. Statistical Analysis

The Statistical Package for the Social Sciences (SPSS) and Microsoft Excel were used to analyze experimental data (SPSS version 20.0, IBM, Armonk, NY, USA). All the experiments were run in triplicates unless otherwise stated. A one-way analysis of variance (ANOVA) and student’s *t*-test were used as statistical tools to validate the results, and to ensure data variability of data. Tukey’s (post-hoc) test (*p* < 0.05) was performed to compare the mean values as described in the specific figure legends. All data in the figures and tables are expressed as mean ± standard deviation (SD).

## 3. Results and Discussion

### 3.1. Identification of the Isolate by 16S rRNA Sequencing and Phylogeny Analysis

The 16S rRNA gene sequence and blast analysis identified the bacterial isolate as *Klebsiella* spp. (SH-1) with 98% genetic sequence similarity (Figure 2a). The submitted sequences can be found in the NCBI database (accession number MF465571.1). 

### 3.2. Effect of Chromium under Optimum Growth Conditions

The optimum growth condition of the isolate was determined. At 37 °C, the growth rate of the isolate was higher than that of room temperature, although the growth rate started to decline when the temperature was increased up to 44 °C. Though the bacterial isolate SH-1 grow in a wide range of pH (5.5 to 10.0), pH 9.0 was found to be optimum for the growth of the isolate. The ability to grow in the presence of Cr(VI) was also determined, in which isolates SH-1 showed a similar growth curve as its control (without chromium). SH-1 showed greater growth efficiency in the presence of Cr(VI) as compared to the negative control (Figure 2b). 

### 3.3. Isolate Showed Different Chromium (VI) Reduction Capacity in Different Media

This study revealed that the isolate has a very high chromium (Cr) resistance property. Therefore, we hypothesized that this isolate might have Cr reducing ability. Afterward, this hypothesis was tested by conducting a time-course study of Cr(VI) reduction in LB broth containing 100 mg/L of Cr(VI) at the optimum temperature (37 °C) and pH (9.0). It has been reported that the optimal Cr(VI) reduction capacity is directly associated with optimum pH (8.0) for the growth of *Bacillus* isolates [43]. In the present study, we observed that this isolate could reduce Cr(VI) rapidly at 37 °C and pH 9. Isolate SH-1 could reduce Cr(VI) approximately 81.12%, 91.66%, and 95% from the medium after 24, 48, and 72 h, respectively (Figure 2c). The previously reported bacteria showed a lower reduction of Cr(VI) over a relatively long period. For instance, it was reported that with 120 h of incubation, *Alcaligens faecalis* reduced only 70.0%, and *Bacillus* sp. (accession number FM208185.1) reduced only 73.41% of initial 100 mg/L Cr(VI) [39,56]. We found that Cr(VI) reduction also occurred in the controls, which were cell-free (Figure 2c). Components may have caused the reduction in the LB medium since no cells were inoculated into the medium. A study by Shanab et al. found that medium components, along with carbon sources, can interact with toxic metals such as arsenic, cadmium, chromium, cobalt, copper, lead, nickel, mercury, and zinc to give misleading results [57]. In addition, Cr(VI) reduction in minimal salt broth after 48 h of incubation confirmed the efficiency of the SH-1 isolate (Appendix A).

To determine the hexavalent chromium reduction efficiency of the isolate in surface water, the isolate was inoculated in pond water and tannery effluents supplemented with 100 mg/L of Cr(VI) without the addition of any nutrients. The reduction efficiency of the isolate was also compared with the indigenous microorganisms (used as the control) present in the water samples. After 72 h of incubation in pond water, SH-1 showed a 45% reduction, whereas the control showed only 13.52% reduction (Figure 2d). Therefore, the reduction percentage was increased by 31.48% after the addition of the isolate. Using one way ANOVA followed by Tukey’s post-hoc test, we obtained significant differences (*p* < 0.05) in percentage reduction by the isolate compared to the indigenous organisms. Subsequently, in tannery effluents, SH-1 showed a 63.08% reduction after 72 h of incubation (Figure 2e). However, the indigenous organisms present in the effluents exhibited only a 36.65% reduction after 72 h (Figure 2e). Though the reduction was observed by indigenous microorganisms in tannery effluents, it was much lower compared to the study isolate (SH-1). Since SH-1 can itself reduce 26.34% Cr(VI) in the presence of indigenous organisms after 72 h of incubation, it indicates that higher Cr(VI) resistance property may help SH-1 to surpass the indigenous organisms.

### 3.4. Isolates Showed Enhanced Chromate Reductase Activity in the Presence of Cr(VI)

The chromate reductase activity of crude cell extracts was obtained by using proteins collected from cells pre-grown, both in the presence and absence of Cr(VI). The soluble proteins from SH-1, both in the presence and absence of Cr(VI), could reduce Cr(VI), as indicated in Table 1. Protein samples from isolate SH-1 treated with Cr(VI) showed a 72.2% reduction, but protein samples from untreated SH-1 showed a 52% reduction. On the other hand, the inactive proteins of SH-1 (boiled at 100 °C) and proteins of the negative control (*V. cholerae*) failed to show chromate reduction. Therefore, reduction caused by Cr(VI) treated proteins were higher than the reduction of untreated proteins of the isolate (Table 1). Because we used the same amount of protein for every sample, we can deduce that the chromate reductase activity increased with Cr(VI) reduction ability (Table 1). This study suggests that the production and the activity of chromate reducing enzymes might have increased in the presence of Cr(VI). In agreement with the previous studies, Cr(VI) reduction might have been caused by the soluble enzymes present in the crude cell extract [21,58]. 

### 3.5. SH-1 Live Biomass Has Better Biosorption Capacity in Response to Different Conditions 

The biosorption capacity of the dried bacterial biomass (SH-1 live and SH-1 dead) for the removal of Cr(VI) was found as a function of initial metal ion concentration, pH, and time.

#### 3.5.1. Better Biosorption Found at Low pH 

As wastewater containing metal ions has different pH values, the biosorption study of Cr(VI) was performed at different pH. Initial pH values were set from 1 to 8 for 100 mg/L Cr(VI) solution. As illustrated in Figure 3a, the capacity of adsorption increased from pH 1.0 to pH 2.0 and then decreased with the increase of pH. We observed a negative correlation of Cr(VI) adsorption individually between the variables (SH-1 live and SH-1 dead) and the increasing pH (r_1_ = −0.89 and r_2_ = −0.94, respectively), which were statistically significant (*p* < 0.01). The maximum chromium biosorption was found to be 33.28 mg/g of dried SH-1 live biomass at pH 2 (Figure 3a). Similar results were observed for the Cr (VI) biosorption by *Ochrobactrum anthropic* and green algae, of which the maximal biosorption capacity was found at pH 2.0 [59,60]. Biosorption depends on the types of adsorbate (metals) and adsorbents (cells of the biomass) [61]. Again, the stability of different forms of Cr(VI) (HCrO_4_^−^, Cr_2_O_7_^2−^, CrO_4_^−^) is mainly dependent on the pH of the medium, where HCrO_4_^−^ is the dominant form at low pH [54]. Therefore, during acidic conditions, functional groups (included amino, carboxyl, hydroxyl, and carbonyl groups) of biosorbent may become protonated and the negatively charged Cr species bind through electrostatic attraction force to the positively charged ions on the surface of the biosorbents [62,63]. Another Cr adsorption study by Sargassum biomass suggested that Cr(VI) is completely reduced to Cr(III) at a low pH, where a part of Cr(III) is adsorbed to the algal biomass [62,64,65]. On the other hand, within a pH of 3.0 to 8.0, biosorption showed no significant differences. High pH causes deprotonation of biosorbent and functional groups become negatively charged, which prevents negatively charged Cr to bind with it [62].

#### 3.5.2. Biosorption Increased with the Increasing Initial Concentration of Cr(VI) 

The biosorption process was conducted with optimized pH (pH 2 for live and pH 1 for dead biomass) at different initial concentrations (20 mg/L to 200 mg/L). When the initial Cr(VI) concentration increased from 20 to 200 mg/L, the absorbing capacity also increased from 8.76 to 51.25 mg/g (dried SH-1 live) and 6.21 to 29.03 mg/g (dried SH-1 dead) biomass (Figure 3b). We observed a positive correlation of Cr(VI) adsorption individually between the variables (SH-1 live and SH-1 dead) and the increasing Cr(VI) concentration (r_1_ = 0.99 and r_2_ = 0.96, respectively), which were statistically significant (*p* < 0.01). The maximum biosorption capacity was found in the dried SH-1 live biomass at 200 mg/L. Similar trends were found in the biosorption study of the *Pantoea* sp. TEM18 biomass [44]. The biosorption capacities of both the live and dead dried biomass were found to have increased with the increasing concentration of Cr ion. This increase may be due to the increase in the competition for the close-fitted binding sites on the surface of biosorbents [61], which indicates the properties of biosorbents (e.g., functional groups, surface area, etc.) and the properties of metal sorbates (e.g., atomic weight, ionic size, etc.) may play an important role in the biosorption process [44].

#### 3.5.3. Effect of Time on Biosorption Process

Figure 3c illustrates the contact time of Cr(VI) by the live and dead dried biomass at a concentration of 200 mg/L at the optimized pH mentioned above. Higher adsorption capacity was found in live biomass than the dead biomass (Figure 3c). Adsorption by the biomass (SH-1 live and SH-1 dead) was increased rapidly throughout the first 60 min and continued to be constant until 90 min. Here, a positive correlation of Cr(VI) adsorption was found between the biomass (SH-1 live and SH-1 dead) and the time up to 90 min (r_1_ = 0.94 and r_2_ = 0.86, respectively), which were statistically significant (*p* < 0.01). Subsequently, after acquiring the equilibrium period at 120 min, no significant differences of Cr(VI) adsorption with time was observed, though the experiments were performed until 360 min. The biosorption equilibrium of Cr(VI) was modeled according to the two adsorption type isotherms (Langmuir and Freundlich).

#### 3.5.4. Adsorption Isotherm

To illustrate the mathematical expression of the biosorption, Freundlich and Langmuir adsorption models were used for dried SH-1 live biomass only. The constants of both isotherms were assessed to compare the capacity of Cr(VI) biosorption by this biomass. 

The values of Q_o_ (maximum monolayer coverage capacity) and K_L_ (Langmuir isotherm constant) were calculated from the slope and the intercept of the Langmuir plot of 1/Q_e_ (Q_e_ = amount of Cr(VI) adsorbed per gram of biomass at equilibrium) versus 1/C_e_ (C_e_ = the equilibrium concentration of Cr(VI)) (Figure 3d). From the data shown in Table 2, the Langmuir isotherm model generated Q_o_ of 72.99 mg/g while our live biomass of SH-1 showed 51.25 mg/g biosorption at 200 mg/L concentration. This study showed a higher adsorption coefficient value (R^2^ > 0.99) for the Freundlich isotherm than the Langmuir adsorption model (R^2^ = 0.94) (Figure 3d,e). This value indicates that the Freundlich model performs better than the Langmuir model for the adsorption of live bacterial biomass. This data also reveals that adsorption in living cell biomass obeys the multi-layer adsorption assumption [45]. Moreover, the Freundlich model showed the value of *n* (*n* = adsorption intensity) was 1.30 (Table 2), where *n* > 1 indicates strong interactions between Cr(VI) and the biosorbents [51,66].

#### 3.5.5. Kinetic Modeling

During the investigation of first pseudo order kinetics, the values of q_e_ and K_1_ were found from the plot of ln(q_e_–q_t_) vs. t (Appendix A). On the other hand, the slope and intercept of plot t/q_t_ vs. t gives the q_e_ and K_2_ values of the second pseudo order model. The first pseudo order kinetic model showed very low linear regression correlation coefficients (*R*^2^ = 0.64 for live and *R*^2^ = 0.53 for dead) and which suggests that the Cr biosorption data onto both biomass did not agree with the pseudo first order model (Appendix A). On the contrary, the calculated values of q_e_ from pseudo-second order kinetics followed the experimental values of q_e_ with coefficient (*R*^2^) values > 0.99 for both of the live and dead biomass, respectively (Figure 3f and Appendix A). These results indicate that the adsorption of Cr(VI) on the bacterial biomass follows pseudo-second order kinetics. A similar trend for pseudo second order kinetics was observed during the biosorption of Cr on both live and dead cells [61]. Kumar and Karthika also concluded that the second order equation fitted well with the adsorption studies [67].

### 3.6. SEM and EDX Analysis

The SEM analysis reveals that the untreated bacterial biomass showed a smooth cell surface, whereas a rough surface, along with surface depression was observed in the Cr(VI) treated biomass (Figure 4a,b). The cell size of the treated biomass was also increased in comparison to that of untreated biomass (Figure 4a,b). These changes might be due to the adsorption of chromium on the biomass cell surface. On top of that, EDX analysis gave some peaks of chromium (with 6.02 mass% analysis, Table 3) in the case of biomass treated with Cr(VI). On the other hand, no such peaks were found from untreated biomass (Appendix A), possibly due to the complexation of chromium species with the biomass cell surface molecules [62]. Additionally, Table 3 showed the elemental content in bacterial biomass after treated with 100 mg/L Cr(VI), which might participate in surface localization of chromium on the biomass cell [68].

### 3.7. FTIR Analysis

FTIR spectroscopy analysis was carried out to obtain the characteristics of the functional groups and to identify the chemical bonds that played a significant role in the process of biosorption of hexavalent chromium. FTIR analysis of chromium exposed and unexposed biomass contained some absorption bands at different wavelengths, which indicate the presence of various functional groups such as an amine (-NH_2_), bonded and non-bonded hydroxyl (O-H) groups, amide (-CONH-), carboxyl (-COOH), aromatic (C = C) ring, alkenes (C = C), aliphatic (-CH_2_) groups, etc. (Figure 4c,d). The peak intensities were slightly different for the chromium treated than that of untreated biomass. In parallel, a slight shifting of some peaks was also confirmed. Strong absorption bands were observed at 3299.65 cm^−1^ for unexposed biomass, which points toward the presence of amine N-H stretching or the presence of bonded or the non-bonded hydroxyl group (O-H) in that region.

On the other hand, a modified vibrational broad band range from 3200 to 3600 cm^−1^ (no sharp peak) was observed in Cr exposed biomass, which might be due to the overlapped stretching of amines and hydroxyl groups or due to the interaction of chromium ion with those functional groups [61,69]. In both Cr exposed and unexposed biomass, the absorption peak of aliphatic (-CH_2_) groups at 2925 cm^−1^ were unchanged, which indicates the asymmetric stretching of proteins, lipids, nucleic acids, and polysaccharides in the biomass cell wall [70]. The strong band at 1068 cm^−1^ for C-O stretching remained the same for both of the Cr exposed and unexposed biomass [71]. For Cr unexposed biomass, other noticeable peaks were also observed at 1654.93, 1542.09, 1450.89, 1399.70, and 1240.02 cm^−1^, indicating the presence of amide and carboxyl groups and aromatic ring on the cell surface [70]. In contrast, Cr exposed biomass showed slight shifting of absorption peaks from 1654.93 to 1658.96, 1542.09 to 1538.58, 1450.89 to 1453.30, and 1240.02 to 1234.45 cm^−1^, indicating the interaction of chromium in the cell surface. Therefore, FTIR analysis of Cr exposed bacterial biomass revealed that all the changes found were may be due to the interaction between organic functional groups (i.e., amino, hydroxyl, and carboxyl groups) and chromium, chelating on the biomass cell surface [72].

### 3.8. Phytotoxicity Assay Using Chickpea Seed Germination

Heavy metals (Cr and Cu) could significantly inhibit the plant and seedling growth [73,74]. Therefore, the toxicity of hexavalent chromium on seed germination along with the growth of roots and shoots was examined in this study. As shown in Figure 5a,b, growth of shoot and root length gradually decreased with increasing concentration of Cr(VI). Calculating the correlation of Cr(VI) concentration individually with the shoot and root length, we observed a negative correlation (r_1_ = −0.74 and r_2_ = −0.91, respectively). It indicates that Cr(VI) affected the growth of shoots and roots of chickpea seeds, although Cr(VI) did not affect the rate of seed germination. Above 50 mg/L of Cr(VI), seedling growth was greatly hampered and no seedling was observed at 100 mg/L (Figure 5a,b). A similar test was conducted to assess the mitigation of toxicity of bacterial treated and untreated Cr(VI) samples (Figure 5c). In control (without Cr), the mean shoot and root length were observed as 14.8 cm and 8.6 cm, respectively (Figure 5d); while the mean shoot and root lengths were 0.7 cm and 0.8 cm, respectively, in the case of 50 mg/L Cr(VI).

Alternatively, the mean shoot and root length of the seeds with Cr(VI) sample that was treated with SH-1 were 10.1 cm and 3.2 cm. Subsequently, the mean shoot and root length of Cr(VI) samples that were treated with dried SH-1 live biomass was 6.7 cm and 3.6 cm, respectively. These results indicate that 50 mg/L Cr(VI) treatment significantly reduces the growth of shoot and root (95.3% and 89.6%, respectively). Of note, Cr(VI) sample that was treated with SH-1 significantly recovered 93% and 75% shoot and root length, respectively, as compared to the 50 mg/L Cr(VI) sample. Similarly, the SH-1 live biomass treated Cr(VI) sample also significantly recovered 89% and 77% shoot and root length, respectively, as compared to the 50 mg/L Cr(VI) (Figure 5c,d). Similarly, the phytotoxicity test by *T. lixii* CR700 and *A. flavus* CR500 revealed the successful detoxification of Cr(VI) [74,75]. This evidence carries significant environmental benefits of using eco-friendly microorganisms to reduce the toxicity of Cr(VI) by either reduction or biosorption.

## 4. Conclusions

The current study gives evidence that hexavalent chromium can be removed by SH-1 cells in LB broth, in tannery effluents, and in pond water. In addition, biosorption studies illustrated the effective sorption capacity of the dried SH-1 live biomass with a high recapture of Cr(VI) in a relatively short time comparing to our Cr(VI) removal process by SH-1 cells. To apply this process from lab to industry, further studies, including chemical modification of the biosorbents along with optimization of the biosorption parameters, might be needed [62]. Moreover, their (SH-1 cells, dried SH-1 live, and dead biomass) beneficial effect on chickpea seedlings rendered a better possibility of successfully applying this chromium resistant bacteria for efficient bioremediation of toxic Cr(VI) from tannery effluents. These findings can be translated into technology by small-scale piloting and, eventually, on a large scale to clean up chrome-polluted wastes, including tannery effluents before discharging the wastes into the environment.

## Figures and Tables

**Figure 1 ijerph-17-06013-f001:**
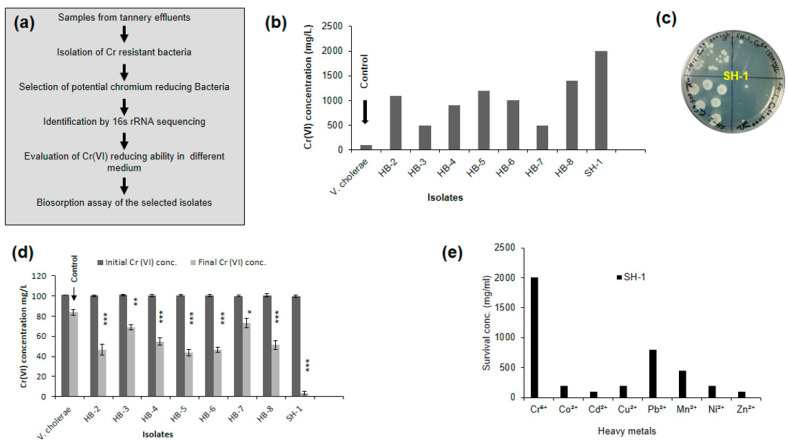
Screening of Cr(VI) reducing bacteria from tannery effluent. (**a**) Schematic diagram of Cr(VI) resistant organism isolation from tannery effluent. (**b**) The resistance of the isolates at different concentrations of Cr(VI). (**c**) Representative growth plate of SH-1 isolate (Top image) from Cr(VI) resistance experiment. (**d**) Reduction of Cr(VI) by the isolates in Luria–Bertani (LB) broth with an initial concentration of 100 mg/L after 72 h of incubation at 37 °C. One way ANOVA followed by Tukey’s post-hoc test was performed to find statistical significance. “*”denotes significant difference between negative control (*V. cholerae*) and other isolates (* *p* <0.05, ** *p* < 0.01, *** *p* < 0.001). (**e**) Resistance to different heavy metals (This experiment was done once).

**Figure 2 ijerph-17-06013-f002:**
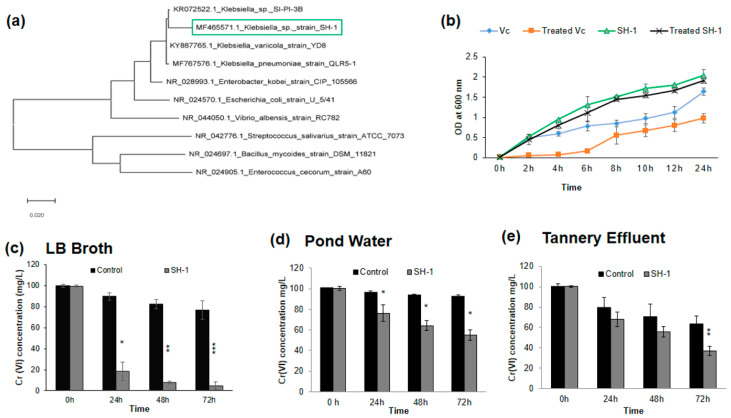
Reduction of Cr(VI) by the isolate SH-1. (**a**) Phylogenetic tree showing the relationship of isolate SH-1 with closely related neighboring sp. (**b**) Effect of Cr(VI) on the growth of SH-1. (**c**) Reduction of Cr(VI) by the isolate in LB broth with an initial concentration of 100 mg/L after 24, 48, and 72 h of incubation at 37 °C. (**d**) Reduction of Cr(VI) by the isolate in pond water with an initial concentration of 100 mg/L after 24, 48 and 72 h of incubation at 37 °C. (**e**) Reduction of Cr(VI) by the isolate in tannery effluents with an initial concentration of 100 mg/L after 24, 48, and 72 h of incubation at 37 °C. Here, LB, PW, and TE denote experiments conducted in Luria–Bertani media, pond water, and tannery effluent, respectively. In Figure 2c–e, paired two-tailed students *t*-test was performed between control and the study group and data are expressed as mean ± standard deviation (SD) (* *p* <0.05, ** *p* < 0.01, *** *p* < 0.001).

**Figure 3 ijerph-17-06013-f003:**
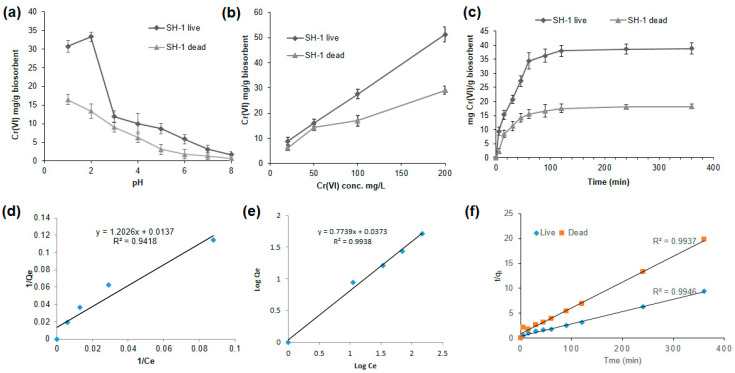
Biosorption assay of isolates, adsorption model, and kinetics. (**a**) Effect of pH and (**b**) initial concentration of Cr(VI) on biosorption process. (**c**) Effect of contact time on biosorption process. (**d**) Langmuir isotherm model and (**e**) Freundlich isotherm model for SH-1 live biomass. (**f**) Pseudo-second-order kinetics plots for biosorption of Cr(VI) by live and dead bacterial biomass.

**Figure 4 ijerph-17-06013-f004:**
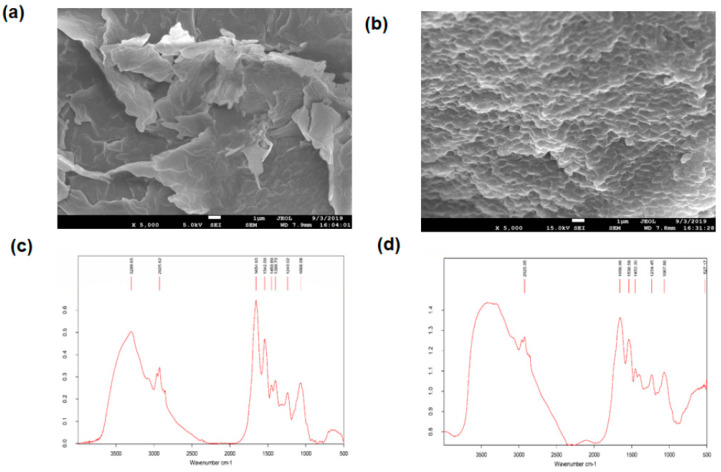
Scanning Electron Microscopy (SEM) and Fourier Transform Infrared Spectrometric (FTIR) analysis of biomass of *Klebsiella sp.* (SH-1). (**a**) SEM of controlled biomass and (**b**) chromium treated biomass. (**c**) FTIR of controlled biomass and (**d**) chromium treated biomass.

**Figure 5 ijerph-17-06013-f005:**
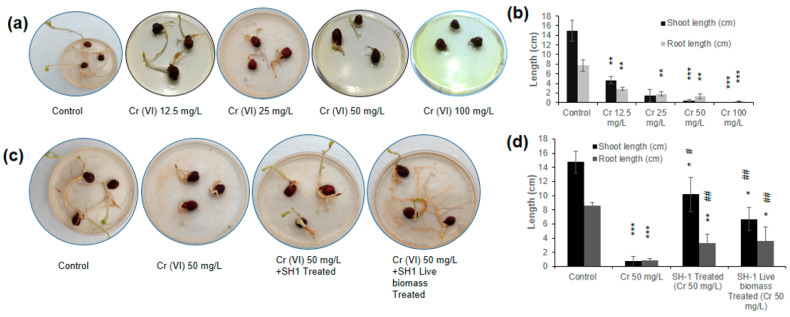
Phytotoxicity test using chickpea seeds. (**a**) Determination of toxicity of different concentrations of Cr(VI) on seed germination and growth. (**b**) The mean shoot and root length of the seeds of Figure 5a. (**c**) Images of reduction of the toxicity of Cr(VI) by bacterial treated samples. (**d**) The mean shoot and root length of the treated seeds of Figure 5c. In Figure 5b,d, paired two-tailed students *t*-test was performed between control and the study group and data are expressed as mean ± standard deviation (SD). “*****” denotes significant differences between control group and study group (*****
*p* < 0.05, ******
*p* < 0.01, *******
*p* < 0.001). “#” indicates significant differences between Cr(VI) 50 mg/L and study group (# *p* < 0.05, ## *p* < 0.01).

**Table 1 ijerph-17-06013-t001:** Cr (VI) reductase activity of cell-free extracts of the isolate SH-1.

Protein Sample	Growth Condition			
	Medium Minus Cr (VI)		Medium Plus 10 mg/L Cr (VI)	
	% Reduction *^a^*	Reductase Activity, U/mg Protein *^b^*	% Reduction *^a^*	Reductase Activity, U/mg Protein *^b^*
SH-1	52 ± 1.52	0.098 ± 0.005	72.2 ± 0.93	0.124 ± 0. 01
*V. cholerae*	3.5 ± 0.8	0.009 ± 0.0005	8 ± 0.42	0.022 ± 0.004

***^a^*** The amount of enzyme that convert 1.0 µM Cr (VI) per min at 37 °C was defined as one unit of Cr (VI) reductase activity. ***^b^*** Cr (VI) reduction was measured after 12 h of incubation. All data in the table are expressed as mean ± standard deviation (SD).

**Table 2 ijerph-17-06013-t002:** Isotherm parameters for Cr(VI) adsorption on the dried SH-1 Live biomass.

Langmuir Isotherm				Freundlich Isotherm			
Q_o_ (mg/g)	K_L_ (L/mg)	R_L_	R^2^	1/*n*	*n*	K_f_ (mg^1 −1/n^ g ^−1^ L^1/n^)	R^2^
72.99	0.01	0.23	0.94	0.77	1.30	1.08	0.99

**Table 3 ijerph-17-06013-t003:** Elemental content in bacterial biomass after treated with 100 mg/L Cr(VI).

Element	Mass (%)	Atom (%)
C K	57.59	75.87
O K	18.87	18.66
S K	2.01	0.99
Cl K	3.80	1.70
Cr K	6.02	1.83
Pt M	11.72	0.95
Total	100.00	100.00

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
