# Peer review of "Bioremediation of Hexavalent Chromium by Chromium Resistant Bacteria Reduces Phytotoxicity"

_ijerph, 2020, doi:10.3390/ijerph17176013_

Round 1
Reviewer 1 Report
It's a good topic to isolate funciontal species to reduce and treat Cr6+ for water protection. The exeprimental design is complete and explanation is good. However, there are some proints need further explanation and revision.
I think Figure 1 is missing (at least fig1 and fig2 are totally the same). This made me quite difficult to catch the details. e.g., what's vc and treated vc in figure 2b, might be V. cholerae as control?
In figure 2, it is quite confused that Cr6+ greatly reduced for 20-40% in control samples. It is possible for the reduction due to the proteins and other agents in the broth. If so, minumum broth or only PBS could be used to verify the performance of the isolates.
In figure 3, it is interesting that SH-1 live cells performed better in pH=2. But, the results are conflict to the fact of cultivation conditions at pH =9 in broth? Moreover, this pH state is very rare in actual state, usually the bulk pH is about 7, in which range both SH1-live and SH-1 dead cells showed nearly zero capacity for adsorption as shown in Figure 3a.
In case of modelling, the isothermal models are popular in the field, and their introduction of background and formular could be reduced. Cause the adsorption in living cell would not obey the single layer adsorption assumption, it is natural that Freucdlich model performs better than Langmiur model. however, the kinetics (1st, saturated, or 2nd order? ) would be helpful for process design and operation, which should be supplied in experimental design. Data of Figure 3c could derive the kinetics.
In which scenario could the Phytotoxicity assat result be used? if some discussion of application, or cross-checking of reference is supplied, it would be helpful for readers to understand the helpfullness of the study.
Author Response
Reviewer’s Response
It's a good topic to isolate functional species to reduce and treat Cr6+ for water protection. The experimental design is complete and explanation is good. However, there are some points need further explanation and revision.
Response:
The authors thank the reviewer for finding this work interesting and also finding our article clear to understand. They also sincerely address the reviewer's suggestions.
I think Figure 1 is missing (at least fig1 and fig 2 are totally the same). This made me quite difficult to catch the details. e.g., what's vc and treated vc in Figure 2b, might be V. cholerae as control?
Response:
Actually, Figure 1 is not missing, and Figure 1 and Figure 2 are not the same as Figure 1 showed the screening process of the Cr(VI) reducing bacteria (Figure 1. Screening of Cr(VI) reducing bacteria from tannery effluent) and Figure 2 showed the reducing capability of the selected isolate SH-1 (Figure 2. Reduction of Cr(VI) by the isolate SH-1). To clarify this issue, the reviewer may also check the Figure legends of these two Figures (Page 37-38, Line 821-840). Here, we denoted VC as V. cholerae; this bacteria was used as a control in this study.
In figure 2, it is quite confused that Cr6+ greatly reduced for 20-40% in control samples. It is possible for the reduction due to the proteins and other agents in the broth. If so, minimum broth or only PBS could be used to verify the performance of the isolates.
Response:
The reduction of Cr(VI) in the controls denoted as LB broth (Figure 2c), Pond water (Figure 2d), and Tannery effluent (Figure 2e); might be due to the presence of proteins and other chemical agents. The authors address the reviewer's concern regarding the performance of the isolates, and an experiment with minimal salt broth is conducted. We added these data in the text (Page 9, Line 198-199 and Page 18, Line 393-394), including figure (Supplementary Figure S6).
In figure 3, it is interesting that SH-1 live cells performed better in pH=2. But, the results are conflict to the fact of cultivation conditions at pH =9 in broth? Moreover, this pH state is very rare in actual state, usually the bulk pH is about 7, in which range both SH1-live and SH-1 dead cells showed nearly zero capacity for adsorption as shown in Figure 3a.
Response:
The authors have used SH-1 bacterial cultures for the reduction study at pH 9 (Page 18, Line 380), Live and Dead SH-1 dried biomass for the biosorption study at pH 1 to pH 8 (Page 20, Line 429).
Though the culture of SH-1 bacterial cells could reduce Cr(VI) at pH 7, the authors got the maximum reduction data at pH 9, and so the authors provided this Cr(VI) reduction data here (Figure 2c, 2d, 2e). Therefore, a noticeable Cr(VI) reduction by the SH-1 bacterial cells could also be possible at pH 7 (bulk pH/ actual state pH).
On the other hand, in the case of the dried SH-1 live and dead biomass of the isolate performed better at pH 2 (Figure 3a). The authors also observed similar trends of maximum absorption capacity at pH 2 to pH 3 in some previous studies-
(Ozdemir, G.; Ozturk, T.; Ceyhan, N.; Isler, R.; Cosar, T. Heavy metal biosorption by biomass of Ochrobactrum anthropi producing exopolysaccharide in activated sludge. Bioresour. Technol. 2003, 90, 71–74)
(Dadrasnia, A.; Chuan Wei, K.S.; Shahsavari, N.; Azirun, M.S.; Ismail, S. Biosorption potential of Bacillus salmalaya strain 139SI for removal of Cr (VI) from aqueous solution. Int. J. Environ. Res. Public Health 2015, 12, 15321–15338).
So, we can use only SH-1 bacterial cells in the treatment plant at pH 7 to pH 9. However, using dried bacterial biomass is quite difficult at actual state pH, the authors give the actual laboratory findings to support the capability of this isolate to remove toxic Cr(VI). To apply the biosorption process from lab to industry, further studies, including chemical modification of biosorbents along with optimization of the biosorption parameters, might be helpful.
In case of modelling, the isothermal models are popular in the field, and their introduction of background and formular could be reduced. Cause the adsorption in living cell would not obey the single layer adsorption assumption, it is natural that Freucdlich model performs better than Langmiur model. However, the kinetics (1st, saturated, or 2nd order?) would be helpful for process design and operation, which should be supplied in experimental design. Data of Figure 3c could derive the kinetics.
Response:
In this study, the authors have found that the Freundlich model performs better than Langmuir model. The authors also derived the kinetics of the biosorption study, which was provided in the text (Page 13, Line 273-283 and Page 22-23, Line 491-504) along with the Figure 3f and Supplementary Figure S7 and Table S8.
In which scenario could the Phytotoxicity assay result be used? if some discussion of application, or cross-checking of reference is supplied, it would be helpful for readers to understand the helpfulness of the study.
Response:
Phytotoxicity assay showed Cr(VI) detoxification capacity of this bacteria, which could be helpful for the plant or seedling growth. According to the reviewer's suggestion, the authors also added some related articles in the text for a better understanding of the reader (Page 25, Line 546-547 and Page 26, Line 567-568).

Reviewer 2 Report
The present review focuses on the investigation of the efficiency of reduction and biosorption of Cr(VI) by chromate resistant bacteria isolated from tannery effluent. The theme is current and can give visibility to the paper. On one hand, the biological part of the paper is well written. On the other hand, the part of the adsorption essays still needs more precision. I made the comments and suggestions below to improve the quality of the document.
Page 3, line 36: The term "Heavy metals" has been considered meaningless. The authors can use "potentially toxic metals", as recommended by the literature.
Page 3, line 37: The authors should avoid such type of "generalization". For example, Cr(III) has an important biological role in the human body. The high toxic form of chromium is Cr(VI).
Page 3, line 47: What form of chromium do the authors talk about? From the tanneries waste, the most common form of chromium is Cr(III) and not Cr(VI). Of course, Cr(III) readily oxidizes to Cr(VI) when typical oxidation conditions are present, but the authors must make this clear to the journal readership. Please, see https://doi.org/10.1016/j.scitotenv.2013.05.004
Page 4, line 59: Among the new technologies, the authors should cite and address a very recent study with core-shell nanoparticles for adsorption and magnetic separation to remove chromium (https://doi.org/10.1016/j.jhazmat.2018.09.008)
Page 6, line 127: The determination of Cr(VI) concentration is a central issue in the survey. The authors have used a modified diphenylcarbazide method, ok. But they should provide the figure of merits for that (limit of detection, limit of quantification, calibration sensitivity, correlation coefficients etc.)
Page 11, line 230: The authors have used the linear forms of the adsorption models in the fittings. However, the nonlinear methods should be applied to obtain parameters that are more accurate than those obtained using the linear method. I strongly recommend the reading of the review written by Tran et al: https://doi.org/10.1016/j.watres.2017.04.014
Page 19, line 389: I think the authors meant “arsenic” instead of “arsenate”, don’t they?
Page 21, line 430: The optimal Cr(VI) biosorption occurs at pH 2. However, most of the investigated effluents have pH in the range of 6-9, where the efficiency of the biosorbent is too low. can this make the use of these materials unfeasible in real life conditions?
Page 21, line 440: In low concentrations (<1 g/L), the prevalent form of Cr(VI) is HCrO4-. The authors should comment on that.
Page 21, line 442: The authors should comment on the type of functional groups of the biosorbent to correlate the Cr(VI) adsorption.
Page 22, figure 3: The fitted parameters (3e) should be listed in a table to be adequately highlighted. What are the experimental errors? They should be clearly expressed.
Page 24, line 489: The RL values must be calculated at a different range of initial adsorbate concentration. An isolated RL value does not have any physical meaning in the present context. Moreover, the RL parameter should be used only if the process is adequately described by the Langmuir model. However, according to Fig.3d the fitting of the adsorption data with this model is poor.
Page 24, line 491: I do not agree at all. The authors have used only four points of adsorption data to provide their results. It is too little! Also, the point (0.0) has no meaning! I strongly recommend the application of non-linear forms of the adsorption models using more experimental points.
Page 25, figure 5: The authors should separate the SEM, EDX and FTIR results because the resolution of the graphs (especially FTIR) is too low. The EDX graphs are less important, so they may be put in the SI (Table 2 is sufficient).
Page 27, line 542: I am not so confident about these findings. Normally, Cr(VI) adsorption can be easily detected due to the peak for chromate vibration at around 950 cm-1. Have the authors checked it?
Page 29, line 584: How can the authors stress that the process takes place in a short time if they have not compared it with other related surveys?
Author Response
Reviewer’s Response
The present review focuses on the investigation of the efficiency of reduction and biosorption of Cr(VI) by chromate resistant bacteria isolated from tannery effluent. The theme is current and can give visibility to the paper. On one hand, the biological part of the paper is well written. On the other hand, the part of the adsorption essays still needs more precision. I made the comments and suggestions below to improve the quality of the document.
Response:
The authors thank the reviewer for appreciating the work and sincerely address the reviewer's suggestions.
Page 3, line 36: The term "Heavy metals" has been considered meaningless. The authors can use "potentially toxic metals", as recommended by the literature.
Response:
As the reviewer suggested, the term "Heavy metals" has been replaced by "potentially toxic metals" in the revised manuscript (Page 3, Line 43).
Page 3, line 37: The authors should avoid such type of "generalization". For example, Cr(III) has an important biological role in the human body. The high toxic form of chromium is Cr(VI).
Response:
The authors have tried to be more specific instead of generalized statements as the reviewer suggested and added some references to make the statements more clear (Page 3, Line 44-45, 51-54).
Page 3, line 47: What form of chromium do the authors talk about? From the tanneries waste, the most common form of chromium is Cr(III) and not Cr(VI). Of course, Cr(III) readily oxidizes to Cr(VI) when typical oxidation conditions are present, but the authors must make this clear to the journal readership. Please, see https://doi.org/10.1016/j.scitotenv.2013.05.004
Response:
The authors thank the reviewer for his constructive comments. Here the report talks about total chromium. Though chrome salts (Cr(III)) that are being used as primary tanning agents during lather processing, the transformation of Cr(III) to Cr(VI) occurs through oxidation under suitable oxidation conditions. For making this clear to readers, the authors have made some changes along with references in the text (Page 3, Line 51-54).
Page 4, line 59: Among the new technologies, the authors should cite and address a very recent study with core-shell nanoparticles for adsorption and magnetic separation to remove chromium (https://doi.org/10.1016/j.jhazmat.2018.09.008)
Response:
As per the reviewer's suggestion, the authors incorporated the information and cited the recent reference in the revised manuscript (Page 4, Line 72-74).
Page 6, line 127: The determination of Cr(VI) concentration is a central issue in the survey. The authors have used a modified diphenylcarbazide method, ok. But they should provide the figure of merits for that (limit of detection, limit of quantification, calibration sensitivity, correlation coefficients etc.)
Response:
The authors appreciate the reviewer for specifying this issue. Actually, the authors have used the 1, 5 diphenylcarbazide method for the measurement of Cr(VI) following the method described by Zahoor et al. The word 'modified' was mistakenly chosen which is removed from the text in the revised manuscript (Page 9 Line 198, Page 10 Line 222, Page 11 Line 233, Page 11 Line 235).
Page 11, line 230: The authors have used the linear forms of the adsorption models in the fittings. However, the nonlinear methods should be applied to obtain parameters that are more accurate than those obtained using the linear method. I strongly recommend the reading of the review written by Tran et al: https://doi.org/10.1016/j.watres.2017.04.014
Response:
The authors appreciate the reviewer for suggesting to use this nonlinear adsorption model. Though it could be better if this model is applied in the study, the authors followed the linear adsorption models and the pseudo kinetic studies to fit their experimental data (Page 13, Line 273-283 and Page 22-23, Line 491-504) (Figure 3f and Supplementary Figure S7 and Table S8).
Page 19, line 389: I think the authors meant "arsenic" instead of "arsenate", don't they?
Response:
The authors have corrected the word 'arsenate' to 'arsenic' (Page 18 Line 392).
Page 21, line 430: The optimal Cr(VI) biosorption occurs at pH 2. However, most of the investigated effluents have pH in the range of 6-9, where the efficiency of the biosorbent is too low. can this make the use of these materials unfeasible in real life conditions?
Response:
Though the dried SH-1 live and dead biomass of the isolate performed better at pH 2 (Figure 3a), SH-1 bacterial cell can be used in the treatment plant at pH 7 to pH 9. The authors admit that the removal of Cr(VI) by dried bacterial biomass is quite difficult at actual state pH (pH 7), but the authors provided the laboratory findings to support the capability of this bacterial isolate to remove Cr(VI). Some previous studies also showed similar trends of maximum absorption capacity at pH 2 to pH 3 (Page 20 Line 436-437). To apply the biosorption process from lab to industry, further studies, including chemical modification of the biosorbents along with optimization of the biosorption parameters, might be helpful.
Page 21, line 440: In low concentrations (<1 g/L), the prevalent form of Cr(VI) is HCrO4-. The authors should comment on that.
Response:
The authors commented in the revised manuscript, as suggested by the reviewer, for a better understanding of the readers (Page 20, Line 437-438).
Page 21, line 442: The authors should comment on the type of functional groups of the biosorbent to correlate the Cr(VI) adsorption.
Response:
The authors have incorporated the necessary information in the text regarding the biosorption process for a better understanding of the readers (Page 20, Line 440-441).
Page 22, figure 3: The fitted parameters (3e) should be listed in a table to be adequately highlighted. What are the experimental errors? They should be clearly expressed.
Response:
The authors thankfully accepted the suggestion and made the changes (Page 22, Line 481, Table 2). The authors calculated the experimental error for pseudo-second-order kinetics and provided the data in Supplementary Table S8.
Page 24, line 489: The RL values must be calculated at a different range of initial adsorbate concentration. An isolated RL value does not have any physical meaning in the present context. Moreover, the RL parameter should be used only if the process is adequately described by the Langmuir model. However, according to Fig.3d the fitting of the adsorption data with this model is poor.
Response:
The authors thank the reviewer for this constructive comment. In this study, the Freundlich model fitted well than the Langmuir model, which has been incorporated in the revised manuscript (Page 22, Line 485-487). The authors also agreed with the reviewer that the RL value is less meaningful in this context, and accordingly, the text has been removed from the revised manuscript.
Page 24, line 491: I do not agree at all. The authors have used only four points of adsorption data to provide their results. It is too little! Also, the point (0.0) has no meaning! I strongly recommend the application of non-linear forms of the adsorption models using more experimental points.
Response:
The authors appreciate the reviewer for suggesting to use this nonlinear adsorption model. Though it could be better if this model is applied in the study, the authors followed the linear adsorption models and the pseudo kinetic studies to fit their experimental data (Page 13, Line 273-283 and Page 22-23, Line 491-504) (Figure 3f and Supplementary Figure S7 and Table S8).
Page 25, figure 5: The authors should separate the SEM, EDX and FTIR results because the resolution of the graphs (especially FTIR) is too low. The EDX graphs are less important, so they may be put in the SI (Table 2 is sufficient).
Response:
The authors have placed the EDX graphs from Figure 4 to Supplementary Figure S9a and S9b in the revised manuscript, as suggested by the reviewer (Page 23, Line 512).
Page 27, line 542: I am not so confident about these findings. Normally, Cr(VI) adsorption can be easily detected due to the peak for chromate vibration at around 950 cm-1. Have the authors checked it?
Response:
Although the authors were concerned about the chromate vibration peak at around 950 cm-1 and also replicated the FTIR experiment, no such vibrational peak was found. For our better understanding, we further checked some related articles and observed similar results (we are providing a reference below).
(Kumar, V., & Dwivedi, S. K. (2019). Hexavalent chromium stress response, reduction capability and bioremediation potential of Trichoderma sp. isolated from electroplating wastewater. Ecotoxicology and Environmental Safety, 185, 109734.)
Page 29, line 584: How can the authors stress that the process takes place in a short time if they have not compared it with other related surveys?
Response:
The authors mean that the biosorption studies showed removal of Cr(VI) by the SH-1 live biomass in a relatively short time comparing to the reduction process by SH-1 cells (Page 26, Line 573-576). Regarding the reviewer's concern, the authors have made substantial changes in the conclusion part of the revised manuscript (Page 26, Line 572-580).

Reviewer 3 Report
Authors did the biosorption testing on live and dead biomass separately.
The dead biomass experiment indicates that the removal of Cr(VI) was removed completely by biosorption process. Do you think live biomass experiment indicates removal of Cr(VI) through biosorption plus oxidation?
In reality, dead and live biomass are living together, so why authors did not do a testing for mixed liquor suspended solids (live plus dead biomass).
I have some minor comments below, please address this. Adding the germination test using chick peak seed was good addition in this paper.
- Line 61-62: “have the risk of secondary chemical contamination”, please explain this clearly.
- Line 63-65: “The use of such microorganisms…….” Sentence is not correct, please check grammar.
- Line 94-95: Please revise sentence “formally identified by one of the co-author Md. Shafiqul Islam, Ph.D”. Provide reference, not full name with degree designation, so delete this sentence.
- Figure 3a should be placed as Table, not figure.
- Figure 4c, d, e and f, x axis and y axis are not clearly visible, please provide quality figure.
Author Response
Reviewer’s Response
Authors did the biosorption testing on live and dead biomass separately.
The dead biomass experiment indicates that the removal of Cr(VI) was removed completely by biosorption process. Do you think live biomass experiment indicates removal of Cr(VI) through biosorption plus oxidation?
Response:
The authors thank the reviewer for addressing this important issue.
Experimental data of this study showed a good biosorption capacity of removing Cr(VI) by live biomass than the dead biomass (Figure 3). Removal of Cr(VI) by live cell biomass depends on both intracellular and extracellular biosorption mechanism; whereas, that of dead biomass depends only on the cell surface sorption. This might be one of the possible reasons in favor of live cell biomass. In addition, the partial removal of Cr(VI) by live cell biomass might also be occurred due to the oxidation process.
In reality, dead and live biomass are living together, so why authors did not do a testing for mixed liquor suspended solids (live plus dead biomass).
Response:
In the present study, the authors only focused on the biosorption capacity of live and dead biomass separately. So, they didn't concern about this test during experimental design. But they will consider the reviewer's valuable suggestion for future study.
I have some minor comments below, please address this. Adding the germination test using chick peak seed was good addition in this paper.
Response:
The authors thank the reviewer for appreciating the incorporation of chick-pea germination experiment in this study.
- Line 61-62: "have the risk of secondary chemical contamination", please explain this clearly.
Response:
During the removal of chromium by chemical coagulation with ferrous sulfate (FeSO4), Cr(VI) is reduced to Cr(III) through the oxidation of Fe(II) to Fe(III). This Fe(III) reacts with water to form ferric hydroxides (Fe(OH)3), which can be used as sorption agents for chromium. Here, there is a chance of secondary chemical contamination of that treated water by iron.
- Line 63-65: "The use of such microorganisms……." Sentence is not correct, please check grammar.
Response:
The authors have made the change in the revised manuscript (Page 4 Line 76).
- Line 94-95: Please revise sentence "formally identified by one of the co-author Md. Shafiqul Islam, Ph.D.". Provide reference, not full name with degree designation, so delete this sentence.
Response:
The authors have removed the line "formally identified by one of the co-author Md. Shafiqul Islam, Ph.D." from the text in the revised manuscript according to the reviewer's suggestions (Page 5, Line 106).
- Figure 3a should be placed as Table, not figure.
Response:
The authors appreciated the reviewer for his suggestion. Maybe the reviewer typed Figure "3a" instead of Figure "3e" mistakenly. The authors have made the changes in Figure "3e" in the revised manuscript, according to the reviewer's suggestions (Table 2).
- Figure 4c, d, e and f, x axis and y axis are not clearly visible, please provide quality figure.
Response:
The authors thank the reviewer for bringing it into the light. The authors have made the changes in the figure sections as the reviewer's suggestion (Page 23, Line 513, Supplementary Figure S9a, and S9b).

Round 2
Reviewer 1 Report
The authors addressed most of the concerns from reviewers.
Abstract: Need a first sentence to identify the background or necessity of the study.
Method: sub-titles could be merged to make the paragraph concise and clean. e.g., sampling sites and sample collection, FTIR and SEM as charaterisation, ...
In adsorption section, the transfomation of Langmiur and Fruendlich for solution is common knowledge in text-book and not necessary explained in detail.
Results: Should focus on most important results. the water quality, screening isolates, could be moved to mehod section.
Author Response
Review response
The authors addressed most of the concerns from reviewers.
Response: The authors thank the reviewer for his positive response.
Abstract: Need a first sentence to identify the background or necessity of the study.
Response: The authors thankfully addressed the reviewer’s concern and incorporated a sentence in the abstract of the revised manuscript (Page 2, Line 24-25).
Method: sub-titles could be merged to make the paragraph concise and clean. e.g., sampling sites and sample collection, FTIR and SEM as charaterisation, ...
In adsorption section, the transfomation of Langmiur and Fruendlich for solution is common knowledge in text-book and not necessary explained in detail.
Response: The authors have made changes according to the reviewer’s suggestions (Page 5, Line 110; Page 14, Line 301-302).
Regarding the adsorption section, the authors provided the fundamental equations of two standard isotherm models in the method part to properly understand the biosorption data. As per the suggestion of the reviewer, the detailed explanation has been removed from the method section.
Results: Should focus on most important results. the water quality, screening isolates, could be moved to mehod section.
Response: According to the reviewer’s suggestion, the water quality and screening isolates have been moved from the result section and merged with the methodology section (Page 6, Line 125-132, 134-135; Page 7, Line 139, 144-150, 154-159; Page 9, Line 201-202; Page 10, Line 212-213) to focus on the most relevant results.

Reviewer 2 Report
In this revised version of the manuscript, the authors incorporate new results and references according to the observations that were made so that the article presents a better quality for publication. The questions raised in comments by the reviewers have been properly addressed. I would like to address only two simple issues: i) the unit of KF in table 2 is incorrect (please, correct as mg1-1/n g-1 L1/n) and ii) the authors insist on applying the linearized forms of the kinetic and adsorption models to adjust the experimental results. The transformation of data for linearization can result in modifications of error structure, the introduction of error into the independent variable, and alteration of the weight placed on each data point, which often leads to differences in the fitted parameter values between linear and nonlinear versions of the models. Of course, this does not make it impossible to publish the article, but it undoubtedly diminishes its quality and credibility.
Author Response
Review response
In this revised version of the manuscript, the authors incorporate new results and references according to the observations that were made so that the article presents a better quality for publication. The questions raised in comments by the reviewers have been properly addressed. I would like to address only two simple issues: i) the unit of KF in table 2 is incorrect (please, correct as mg1-1/n g-1 L1/n) and ii) the authors insist on applying the linearized forms of the kinetic and adsorption models to adjust the experimental results. The transformation of data for linearization can result in modifications of error structure, the introduction of error into the independent variable, and alteration of the weight placed on each data point, which often leads to differences in the fitted parameter values between linear and nonlinear versions of the models. Of course, this does not make it impossible to publish the article, but it undoubtedly diminishes its quality and credibility.
Response:
(i) The authors thank the reviewer for finding out the mistake, which has been corrected accordingly (Table 2).
(ii) The authors appreciate the reviewer’s valuable suggestion and will gladly apply the nonlinear model in the future study.
